# A Discretized Overlap Resolution Algorithm (DORA) for resolving spatial overlaps in individual-based models of microbes

Ihab Hashem[1], Jian Wang[1], Jan F.M. Van Impe [iD][1]*

KU Leuven, Chemical Engineering Department, BioTeC & OPTEC, Ghent, Belgium

* jan.vanimpe@kuleuven.be

**Data availability statement:** The code and raw data are openly available at

## Abstract

Individual-based modeling (IbM) is an instrumental tool for simulating spatial microbial growth, with applications in both microbial ecology and biochemical engineering. Unlike Cellular Automata (CA), which use a fixed grid of cells with predefined rules for interactions, IbMs model the individual behaviors of cells, allowing complex population dynamics to emerge. IbMs require more detailed modeling of individual interactions, which introduces significant computational challenges, particularly in resolving spatial overlaps between cells. Traditionally, this is managed using arrays or kd-trees, which require numerous pairwise comparisons and become inefficient as population size increases. To address this bottleneck, we introduce the Discretized Overlap Resolution Algorithm (DORA), which employs a grid-based framework to efficiently manage overlaps. By discretizing the simulation space further and assigning circular cells to specific grid units, DORA transforms the computationally intensive pairwise comparison process into a more efficient grid-based operation. This approach significantly reduces the computational load, particularly in simulations with large cell populations. Our evaluation of DORA, through simulations of microbial colonies and biofilms under varied nutrient conditions, demonstrates its superior computational efficiency and ability to accurately capture microbial growth dynamics compared to conventional methods. DORA's grid-based strategy enables the modeling of densely populated microbial communities within practical computational timeframes, thereby expanding the scope and applicability of individual-based modeling.

## Author summary

In microbial ecology and biochemical engineering, individual-based models (IbMs) are essential for simulating population dynamics at the cellular level. A key challenge is

https://doi.org/10.48804/AIXDNO and are shared under a Creative Commons license.

**Funding:** This work was supported by the Research Foundation Flanders (FWO) (project G0B4121N to JVI), by the European Commission within the framework of the Erasmus+ Programme (FOOD4S Erasmus Mundus Joint Master Degree in Food Systems Engineering, Technology and Business 619864-EPP-1-2020-1-BE-EPPKA1-JMD-MOB to JVI), by the European Union's Horizon 2020 research and innovation programme (under the Marie Skłodowska-Curie grant agreement No. 956126 E-MUSE to JW and JVI) and by the European Union's Horizon Europe research and innovation programme (under Grant Agreement No. 101058422 SUPREME - SUstainable nanoPaRticles Enabled antiMicrobial surfacE coatings - to IH and JVI). The funders had no role in study design, data collection and analysis, decision to publish, or preparation of the manuscript.

**Competing interests:** The authors have declared that no competing interests exist.

resolving spatial overlaps among cells in large-scale simulations. We introduce DORA, an algorithm that translates cell positions into a grid-based occupancy matrix, applies a diffusion-like process to resolve overlaps, and then translates the resulting adjustments back to individual cell movements. This approach reduces computational complexity, making large-scale IbM simulations more feasible.

## Introduction

Individual-based modeling is a bottom-up approach that explicitly simulates each member of a population, allowing for the emergence of collective dynamics from their interactions [1]. Individual-based Models (IbMs) have been pivotal in microbial ecology and biochemical engineering [2]. In microbial ecology, IbMs have facilitated investigations into the evolution of altruistic behavior among bacteria [3], regulation of public goods, the relationship between spatial structure and social interactions [4], and the preservation of biodiversity within biofilms [5]. In biochemical engineering, IbMs are utilized for modeling wastewater treatment processes [6] and designing bioreactors [7]. The strengths of IbMs lie in their capacity to reveal complex system behaviors from simple individual interactions and in their ability to demonstrate the influence of spatial organization on system-level outcomes [8,9]. However, these advantages are tempered by the significant data and computational resources required by the detailed, spatially explicit models [8,9].

Spatial models of microbial systems can be categorized as either on-lattice or off-lattice, depending on whether cells are constrained to a fixed grid or allowed to move freely in continuous space [10]. On-lattice models, such as cellular automata (CA), represent space as discrete grid units, offering computational advantages like efficient neighbor identification and simplified interaction calculations. However, the rigid grid structure can introduce spatial limitations, such as directional biases and challenges in accurately representing irregular cell shapes and movements. In contrast, off-lattice models operate in continuous space, allowing for more precise simulations of cell positioning and interactions [9]. This approach eliminates grid-induced anisotropies and provides greater flexibility in modeling complex biological processes, including dynamic cell shapes and mechanical interactions. Despite these advantages, off-lattice models tend to be more computationally intensive and require sophisticated algorithms for tasks like neighbor searching and collision detection, which can limit their feasibility for large-scale simulations [10].

Simulating microbial growth using IbMs encompasses several computationally demanding tasks: simulating the diffusion of nutrients and other chemical species across a grid, modeling the growth of individual cells, and implementing a cell shoving algorithm to resolve spatial overlaps among neighboring cells [9]. The latter poses the most significant challenge. It has been noted that while modeling the growth of $N$ cells involves simulating Monod kinetics for each cell and is computationally intensive, it ultimately remains an $O(N)$ operation, meaning its computational complexity scales linearly with the number of cells [9]. In contrast, cell shoving requires pairwise comparisons of each cell against all others in the population in order to identify neighboring cells, escalating the computational complexity to $O(N^2)$. This computational burden can be partially alleviated by leveraging spatially organized data structures, such as kd-trees. These structures allow for the recursive encoding of cell positions, enabling efficient partitioning of the simulation space and facilitating rapid search traversal to identify neighboring cells [11]. Consequently, the complexity of locating neighboring cells for the shoving process is reduced to $O(\log N)$ operations per cell, thus reducing the overall complexity to $O(N \log N)$ [8,9].

In this paper, we introduce a novel method for simulating cell shoving and resolving overlaps between neighboring cells through a Discretized Overlap Resolution Algorithm (DORA). DORA models the physical space occupied by cells as a discrete grid, represented by an "occupancy matrix." Within this matrix, the value of each grid unit quantifies its occupancy level, with values exceeding one indicating regions of overlap. The algorithm processes this matrix to generate movement vectors, directing cells away from congested areas, thereby enabling efficient overlap resolution without the need for direct comparisons between individual cells. This innovative approach significantly reduces computational complexity to $O(N)$, enhancing the feasibility of simulating larger population densities that were previously unfeasible with traditional shoving algorithms. The structure of the manuscript is as follows: Materials and methods details the framework of the algorithm. Results and discussion evaluates DORA's performance against the conventional kd-tree-based approach in handling overlaps, through case studies on the growth of colonies and biofilms under various nutrient conditions. This section also discusses the advantages and limitations of each method. Finally, Conclusion provides a summary of the paper's conclusions.

## Materials and methods

Traditionally, managing cell positions and detecting overlaps in IbMs have relied on data structures such as arrays or kd-trees, as depicted in Fig 1A. While arrays offer a straightforward method for storing cell locations, they necessitate pairwise comparisons between each cell and every other cell, leading to a computational complexity of $O(N^2)$ and becoming increasingly inefficient as the population grows. In contrast, kd-trees, a binary tree variant, provide a more sophisticated approach by efficiently partitioning space and facilitating quicker queries for neighboring cells. The name "kd-tree" originates from "k-dimensional tree" as it involves partitioning space into bins at each node using $k$ dimensions, typically using a series of alternating vertical and horizontal cuts. This hierarchical data structure enables recursive space partitioning, significantly reducing the $N$ comparisons required by arrays to $O(\log N)$ operations for each cell query, such as range searches and nearest neighbor searches, thus allowing faster identification of potential overlaps. Despite the efficiency gains per cell query, the overall computational complexity for querying the entire population with a kd-tree still escalates to $O(N \log N)$, marking a substantial improvement over array-based methods but remaining significant for large populations [8,9,11]. This complexity is compounded by the need for periodic rebalancing of the kd-tree to maintain querying efficiency, as cells move and the population dynamics evolve. Rebalancing, depending on the extent of spatial changes, may involve partial to complete reconstruction of the tree structure, adding to the computational and memory overhead. Therefore, while kd-trees offer an advanced solution for managing cell positions and detecting overlaps, especially in high-density scenarios, they present their own set of challenges, including a still significant computational burden, higher memory requirement than arrays and need for tree rebalancing [12].

After identifying the neighbors of a focal cell, resolving overlaps typically involves assessing the extent to which cells intrude upon each other's space. As illustrated in Fig 1B, when the sum of the radii $r_1 + r_2$ of two cells exceeds the distance $d$ between their centers, an overlap is identified. Classical resolution methods might then utilize a 'cell shoving' technique, where the focal cell is displaced to resolve the overlap, simulating a repulsive force [8]. Hence, while resolving an already identified overlap is straightforward, the underlying tasks of storing and searching for cell positions to identify potential overlapping cells pose significant computational challenges, even with advanced data structures like kd-trees.

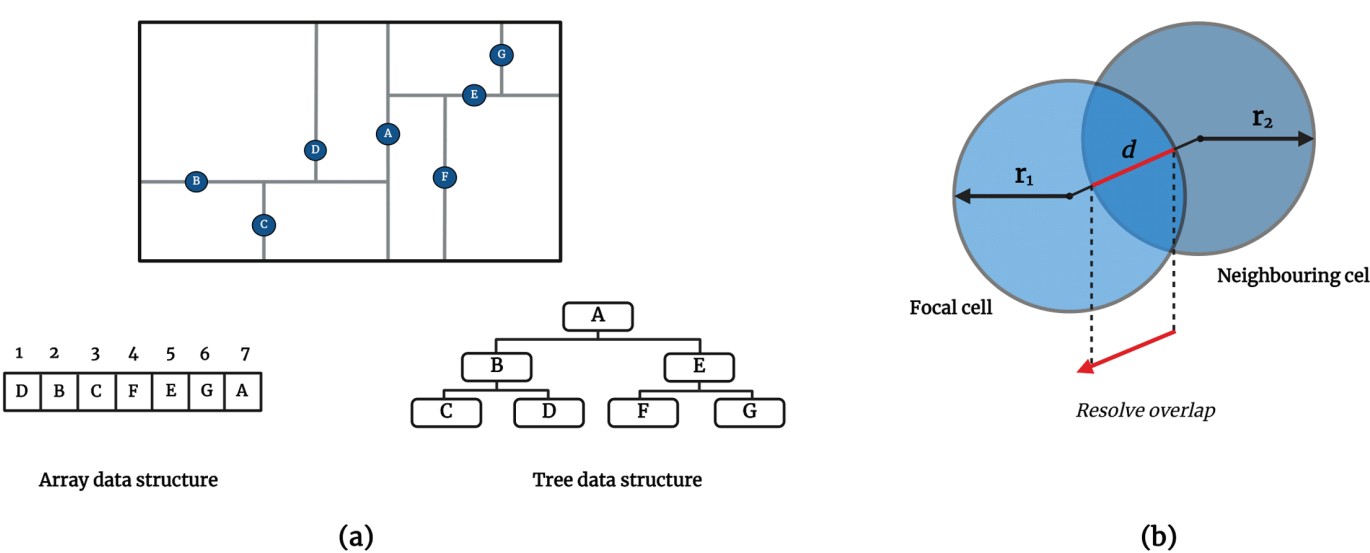

**Fig 1.** (**A**) Comparison of two data structures for tracking cell locations: an array and a kd-tree. The array is a simple linear data structure where cell positions are stored sequentially, requiring direct access to each cell location for overlap detection, which can become computationally expensive for large datasets. The kd-tree, on the other hand, is a hierarchical binary tree structure used to organize points in a k-dimensional space. The construction of a kd-tree involves recursively partitioning the space into two halves along an axis-aligned plane. The process starts by choosing a dimension (e.g., x-axis) and finding the median point along that dimension to serve as the root. Subsequent layers of the tree alternate the dimension used for partitioning (e.g., switching to the y-axis for the next split), and this process continues recursively until all points are organized into leaf nodes. This hierarchical structure allows efficient querying of cell positions and neighbors, reducing the number of comparisons needed to identify overlaps. (**B**) Illustration of overlap calculation between two cells with radii $r_1$ and $r_2$ and the distance $d$ between their centers. Overlap is determined when the sum of the radii exceeds the distance between the centers, indicating a need for resolution through cell shoving.

## Algorithm overview

DORA aims to resolve spatial overlaps between neighboring cells in IbMs of microbial growth through a grid-based framework. This eliminates the need for pairwise comparisons among cells, saving both time and memory. The algorithm consists of three main steps:

1. **Forward translation**: This stage involves the construction of an occupancy matrix that quantifies the extent to which each grid unit is occupied by cells, hence translating spatial information—positions and radii of the cells within the simulation—into occupancy values. Grid units fully occupied by a single cell are assigned an occupancy value of one, while those over-occupied by multiple cells, indicating overlap, have values greater than one.

2. **Overlap resolution**: In this step, a diffusion-like process is applied to the occupancy matrix to resolve overlaps. This process iteratively adjusts the occupancy values, simulating a repulsive force that acts to separate overlapping cells. Concurrently, a four-dimensional motion tensor is maintained, recording the directional displacements required at each grid unit to alleviate overlaps, where each dimension of the tensor corresponds to one of the cardinal directions.

3. **Backward translation**: The final stage entails translating the directional displacements recorded in the motion tensor back to the respective individual cells. For each cell, a movement vector is derived from the displacement values at the grid units it occupies. This vector then dictates the cell's movement in the following simulation step to resolve potential overlaps.

Fig 2 provides an illustrative overview of DORA's flow, while Algorithm 1 outlines the key steps of the algorithm.

**Algorithm 1** Discretized Overlap Resolution Algorithm (DORA)

```
 1: Input: Spatial attributes of cells (positions, radii)
 2: Output: Adjusted positions of cells to resolve overlaps
    Forward Translation
 3: for each cell do
 4:    Compute cell boundaries relative to grid units
 5:    Update occupancy matrix Ω based on cell's spatial extent
 6: end for
    Overlap Resolution
 7: Initialize excess matrix E from Ω
 8: while excess occupancy exists do
 9:    for each grid unit (i,j) do
10:       Compute excess occupancy Eᵢⱼ
11:       Redistribute excess to adjacent units, update Ω
12:       Update motion matrix M based on excess redistribution
13:    end for
14: end while
    Backward translation
15: for each cell do
16:    Compute movement vector from M
17:    Adjust cell position based on movement vector
18: end for
```

## Forward translation

The forward translation step maps the spatial attributes of cells—their locations and sizes—onto an occupancy matrix, denoted as $\Omega$. This matrix discretizes the simulation environment into $H \times W$ units, where $H$ and $W$ denote the height and the width of the environment,

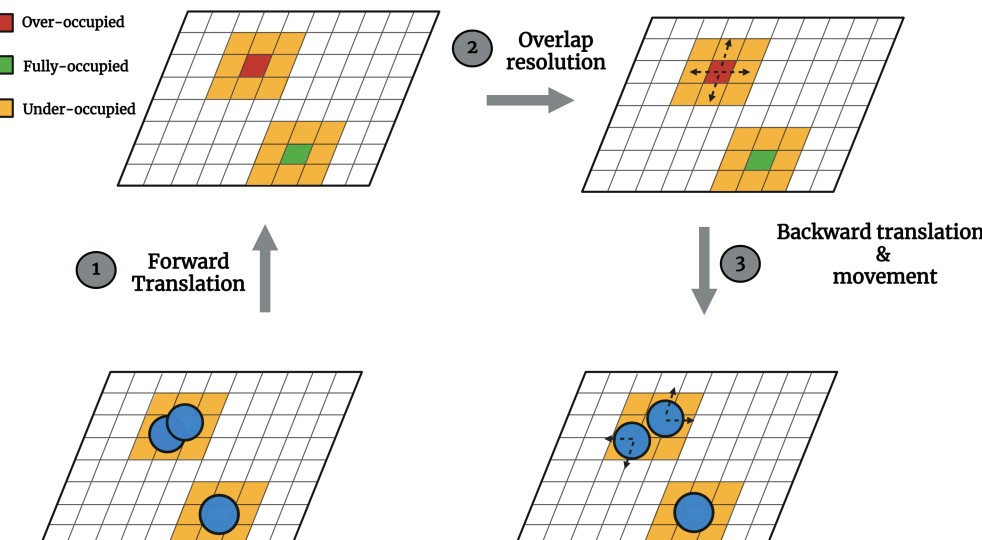

**Fig 2. Conceptual illustration of the Discretized Overlap Resolution Algorithm (DORA).** It shows the three main phases of the algorithm: Forward Translation, where the occupancy matrix is constructed from the spatial attributes of cells; Overlap Resolution, where overlaps are identified and resolved through a diffusion-like solution process of occupancy matrix; and Backward Translation, where the computed displacements are translated back to individual cell movements to achieve a non-overlapping cellular arrangement.

respectively. Each unit stores a value representing the fraction of the grid occupied by bacterial cell(s).

To represent the spatial extent of a cell within the grid, the algorithm begins by computing the boundaries of each cell relative to the grid units. For a cell centered at coordinates $(x_c, y_c)$ with radius $r$, its boundaries along the $X$ and $Y$ dimensions are defined as:

$$x_{\text{left/right}} = x_c \pm \frac{r}{w}, \quad y_{\text{bottom/top}} = y_c \pm \frac{r}{h} \tag{1}$$

where $x_{\text{left/right}}$ and $y_{\text{bottom/top}}$ denote the left/right and bottom/top boundaries of the cell, respectively. Here, $x_c$ and $y_c$ are the coordinates of the cell's center in grid units, $r$ is the cell's radius in $\mu m$, and $w$ and $h$ represent the width and height of a grid unit in $\mu m$ per grid unit. The values of $w$ and $h$ are chosen to be on the same scale as the cell radius, typically around $1\ \mu m$ per grid unit.

Subsequently, for each grid unit indexed by $(i, j)$, the occupancy by a cell $k$, $\Omega_{ij}^{(k)}$, is calculated as the fractional overlap between the grid unit and the cell. This is determined by combining the horizontal and vertical overlaps:

$$\Omega_{ij}^{(k)} = \frac{\text{horizontal overlap} \times \text{vertical overlap}}{w \times h},$$

where $w$ and $h$ are the width and height of a grid unit.

The horizontal overlap is given by:

$$\max(0, \min(i + 1, x_{\text{right}}) - \max(i, x_{\text{left}})),$$

and the vertical overlap is computed similarly:

$$\max(0, \min(j + 1, y_{\text{top}}) - \max(j, y_{\text{bottom}})).$$

The horizontal overlap is determined by $\max(i, x_{\text{left}})$, which identifies where the overlap starts, ensuring it begins only where the grid and cell intersect, and $\min(i + 1, x_{\text{right}})$, which identifies where the overlap ends. Subtracting these values gives the overlap length, with $\max(0, \cdot)$ ensuring no negative values. The vertical overlap is calculated similarly using the cell's top ($y_{\text{top}}$) and bottom ($y_{\text{bottom}}$) boundaries. The product of the horizontal and vertical overlaps gives the total overlap area, which is then normalized by the grid unit's area ($w \times h$) to represent the fractional occupancy of the cell within the grid unit.

The matrix $\Omega$ is then populated with these computed occupancy values for each grid unit, offering a representation of occupancy levels across the grid. A geometric representation of the occupancy calculation process is illustrated in Fig 3A. For a grid unit at indices $(i, j)$, the total occupancy $\Omega[i][j]$ is the sum of the occupancies contributed by each cell $k$ that occupies the unit:

$$\Omega[i][j] = \sum_{k} \Omega_{ij}^{(k)} \tag{2}$$

Values in $\Omega$ less than one indicate under-occupied units, exactly one signifies fully occupied units, and greater than one identifies over-occupied grid units, where more than one cell occupies the grid unit. The subsequent step aims to identify the localized motion vectors necessary to resolve these overlaps.

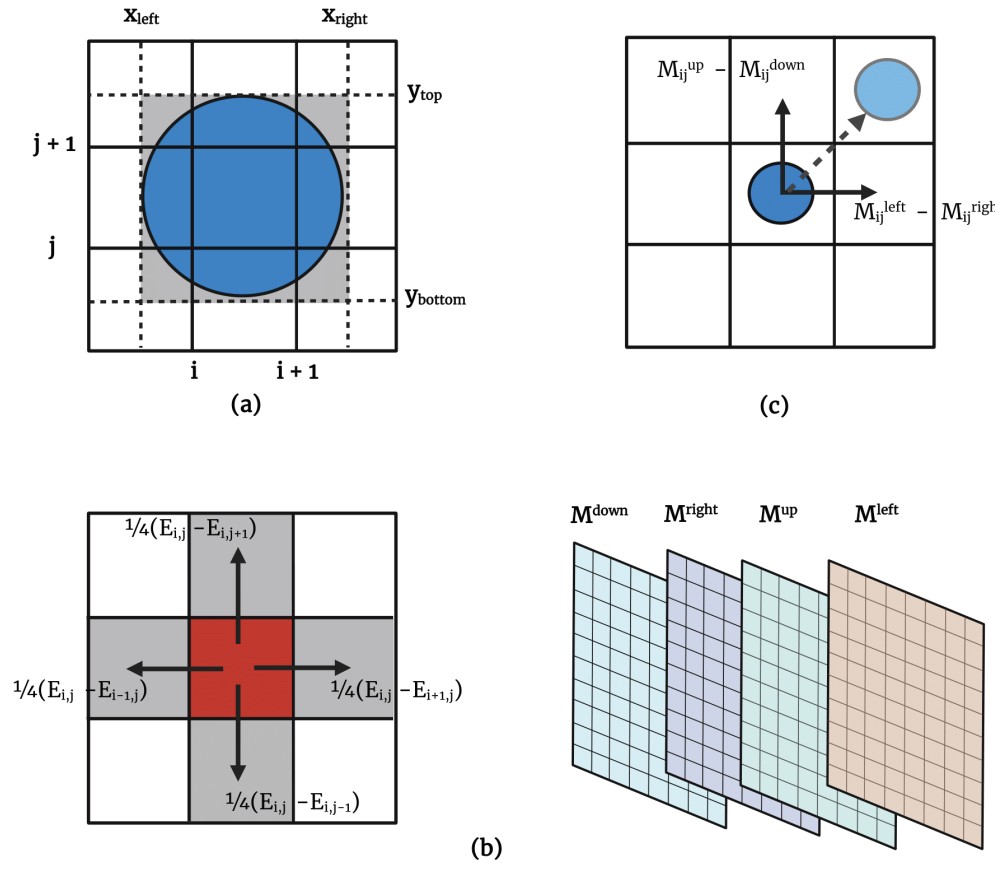

**Fig 3. Illustration of the geometric aspects of the algorithm.** (**A**) Occupancy calculation for a cell, showing the boundaries $x_{left}$, $x_{right}$, $y_{top}$, and $y_{bottom}$ relative to the grid units. (**B**) Redistribution of excess occupancy during the diffusion step, visualized with arrows indicating the directions of movement. The four-dimensional motion tensor, which stores directional displacements for each grid unit in the cardinal directions, is displayed on the right. (**C**) Back-translation and movement, demonstrating how movement vectors are derived from the displacements stored in the motion tensor and applied to adjust the positions of individual cells to resolve overlaps.

## Overlap resolution

The second stage focuses on resolving the overlaps identified in the $\Omega$ matrix. An Excess Matrix, $E$, is defined to capture surplus occupancy, with the excess occupancy for each grid unit $(i,j)$, denoted $E_{ij}$, indicative of overlap, computed as:

$$E_{ij} = \max(0, \Omega_{ij} - 1) \tag{3}$$

The algorithm iteratively redistributes the identified excesses to adjacent grid units to achieve equilibrium, simulating cellular shoving. For a grid unit $(i,j)$, the sum of excess occupancies from the four immediate neighbors of the grid unit $(i,j)$ is computed. Let $\mathcal{N}_{ij}$ denote the Von Neumann neighborhood of the grid unit. The Von Neumann neighborhood of a grid unit includes the four adjacent units in the up, down, left, and right directions.

The total excess occupancy of the neighboring cells, $E_{i,j}^{\mathcal{N}}$, is defined as:

$$E_{i,j}^{\mathcal{N}} = \sum_{(k,l) \in \mathcal{N}_{ij}} E_{kl} \tag{4}$$

To adjust for the movement of excess occupancy, the updated $\Omega'$ matrix is calculated from the following equation:

$$\Omega'_{ij} = \Omega_{ij} - \alpha \cdot \left( E_{ij} - \frac{1}{4} \cdot E_{i,j}^{\mathcal{N}} \right) \tag{5}$$

This redistribution is controlled by a numerical diffusivity factor $\alpha$, to ensure numerical stability. The term $E_{ij} - E_{i,j}^{\mathcal{N}}$ represents a local gradient of excess occupancy, driving the redistribution of cells to neighboring grid units. The diffusion process and the redistribution of excess occupancy are shown in Fig 3B. Iterations proceed until changes in occupancy fall below a designated threshold, signifying a steady state where all excess occupancies have been redistributed. This approach ensures that only grid units with over-occupancy are targeted, distinguishing the process from standard diffusion processes, which aim for a uniform spatial distribution. The motion here is driven solely by the need to alleviate areas of excess occupancy. The $\frac{1}{4}$ term arises from the Von Neumann neighborhood, where each grid unit has four neighboring cells (up, down, left, and right). This ensures that excess occupancy is redistributed evenly across the four adjacent cells.

A four-layered motion tensor, $M$, tracks the net displacements required for overlap resolution. Each layer of $M$ corresponds to one of the cardinal directions: down ($d = 0$), right ($d = 1$), up ($d = 2$), and left ($d = 3$). For a given direction $d$ at grid unit $(i,j)$, and considering a neighbor $(k,l)$ defined per direction, the update rule is:

$$M_{ij}^{(d)} \mathrel{+}= \alpha \cdot \frac{1}{4} \cdot \left( E_{ij} - E_{kl} \right) \tag{6}$$

This scheme captures the directional adjustments necessary for overlap resolution, with the layers of the motion matrix tracking the required displacements of the cells occupying each grid unit.

## Back-translation and movement

In the final phase, the algorithm translates the adjusted occupancy levels from the matrix $\Omega$ back into movement vectors for the cells, using the motion tensor $M$. This process involves calculating the movement vectors for each cell in both horizontal (x-direction) and vertical (y-direction) based on the displacements recorded in $M$. Specifically, for a grid unit at coordinates $(i,j)$ occupied by a single cell, the local movement vector $(v_{i,j}^x, v_{i,j}^y)$ is computed as follows:

$$(v_{i,j}^x, v_{i,j}^y) = (M_{i,j}^{\text{right}} - M_{i,j}^{\text{left}}, M_{i,j}^{\text{top}} - M_{i,j}^{\text{bottom}}) \tag{7}$$

Here, it should be noted that cells not directly involved in overlaps might still need to adjust their positions to help alleviate congestion elsewhere in the simulation. This is a direct outcome of the earlier phases of the algorithm that handle the resolution of overlaps using the occupancy matrix. The back-translation and resulting cell movement are depicted in Fig 3C.

In situations where multiple cells ($N > 1$) occupy a single grid unit, the movement vectors are distributed among the cells based on their relative magnitudes of movement in $M$. Specifically, the proportion of cells moving rightward versus leftward, given by $(M_{i,j}^{\text{right}}/M_{i,j}^{\text{left}})$,

determines how many cells are assigned to move in each direction. Similarly, the vertical movements are distributed based on ($M_{i,j}^{\text{top}}/M_{i,j}^{\text{bottom}}$). Once a direction is assigned, the movement vector magnitude is evenly shared among the cells moving in that direction, ensuring that each cell contributes fairly to the overall movement pattern.

For cells that span multiple grid units, the overall movement vector ($V^x$, $V^y$) is determined by summing the effects from all grid units covered by the cell:

$$V^x = \sum_{(i,j)\in \text{Cell's grid units}} v_{i,j}^x, \quad V^y = \sum_{(i,j)\in \text{Cell's grid units}} v_{i,j}^y \tag{8}$$

Following the computation of these vectors, cells are accordingly repositioned within the simulation domain, thus addressing the overlaps delineated earlier in the process.

Several approximations are employed to manage the computational complexity of the algorithm. Specifically, (i) circular cells are approximated as squares to simplify boundary calculations and overlap detection. In addition, (ii) the simulation environment is discretized into a grid, which limits spatial precision as the fixed grid resolution may not capture finer-scale details. Furthermore, (iii) cells are treated as rigid, non-deformable entities, an assumption also utilized in the kd-tree-based approach to facilitate the overlap resolution process. While this method effectively resolves overlaps for circular cells, it does not take rotational forces into consideration. As a result, simulating cells with more complex shapes, such as rod-shaped cells, would require additional modifications, which are not considered in the current framework.

## Individual-based modeling toolkit

The DORA algorithm, alongside a kd-tree-based method for resolving spatial overlaps among cells, have been integrated into MICRODIMS, an in-house individual-based modeling platform developed for simulating microbial growth [13–15]. Built upon the Repast Simphony toolkit [16], MICRODIMS adheres to the design principles common to IbM toolkits as outlined in the literature [5,8,17,18]. Within this platform, cells are modeled as discrete entities exhibiting Monod growth kinetics, as well as other biological processes such as reproduction and lysis. The kd-tree-based approach employs a relaxation algorithm for overlap resolution [8]. Substrate diffusion follows Fick's law and is numerically resolved using the discretized Forward-Time Central-Space (FTCS) scheme. Colony growth simulations occur within a $600 \times 600 \, \mu m^2$ environment, with periodic boundary conditions on all sides. For biofilm growth experiments, the simulation domain is set to $800 \times 200 \, \mu m^2$, featuring Neumann boundary conditions at the solid bottom edge to represent a no-flux boundary, Dirichlet boundary conditions at the top for an environment with infinite resources, and periodic boundary conditions along the lateral edges.

Biofilm simulations are initiated with 40 cells uniformly distributed at the bottom of the simulation environment, whereas colony simulations start with a single cell at the center. The diffusion processes are simulated with a fast time step of $5 \times 10^{-4}$ min, and the movement of cells, along with other metabolic processes, is simulated every 0.01 min of the simulation. Detailed descriptions of cell growth parameters are provided in the S1 Table and are also available in earlier works [13,14].

## Results and discussion

To assess the performance of DORA, we conducted a comparative analysis with the kd-tree-based method within IbMs of microbial growth developed in MICRODIMS. DORA utilizes

a grid-based discretization of the simulation space, allowing each cell to interact exclusively with the grid. The method updates grid occupancy and derives movement vectors in a manner that only requires solving the grid once per iteration. Consequently, it significantly reduces computational demands by eliminating the need for pairwise cell comparisons. To validate these assertions, we simulated colony and biofilm growth under various nutrient conditions—100 mg/L, 10 mg/L, and 1 mg/L—to elicit a spectrum of growth morphologies, from homogeneous expansions to fractal patterns [19]. These simulations served as a testbed for evaluating DORA's computational efficiency as well as its ability to replicate the complex spatial dynamics of microbial growth under various conditions.

## Colony growth

The initial set of experiments focused on the growth dynamics of a microbial colony, conducting 30 simulations in a nutrient-rich environment with a concentration of 100 mg/L. As illustrated in Fig 4A, the colony exhibits uniform radial expansion and a high growth rate attributable to the abundance of nutrients, achieving a population density of approximately $2 \times 10^4$ cells. This homogenous growth pattern arises predominantly from the minimal nutrient gradients present under these conditions, facilitating uniform cell division along the colony's periphery [2,19].

In these simulations, we utilized a Von Neumann neighborhood to handle overlap resolution. While this neighborhood is anisotropic, as it emphasizes movements along the cardinal directions and underrepresents diagonal movements, the colonies still exhibited nearly circular growth. This suggests that the spatial dynamics of colony expansion were not strongly impacted by the anisotropy of the overlap resolution step. We attribute this to the stochastic elements inherent in the simulation, such as random cell reproduction and placement, which likely averaged out any directional biases introduced by the choice of neighborhood. Additionally, we tested an alternative configuration of the algorithm using a Moore neighborhood (see S1 Algorithm) but found no significant difference in the observed growth patterns or overlap
ratios.

Figs 4B and 4C present a comparative analysis of the computational efficiency between the grid-based DORA algorithm and the kd-tree method, particularly focusing on the overlap resolution step. It is noted that the resolution time—the duration required to resolve cellular overlaps at each long simulation timestep 0.01 min—increases exponentially as the simulation progresses for both algorithms, correlating with the exponential growth in cell numbers. However, DORA consistently outperforms the kd-tree approach in terms of computational efficiency. This advantage is attributed to DORA's strategy of directly mapping spatial data onto a discretized grid, which eliminates the need for computationally intensive pairwise comparisons and neighbor searches, typical of kd-tree algorithms. This efficiency becomes increasingly significant as the cell population surpasses $10^4$, where kd-trees face scalability challenges due to the growing number of potential neighbors per cell [9]. In contrast, DORA maintains relatively stable performance up to cell populations of the order of $10^5$, with memory capacity for storing cellular data in the IbM becoming the primary bottleneck beyond this threshold.

While DORA offers substantial improvements in computational efficiency, it introduces a level of approximation in the overlap resolution process, unlike the exact approach afforded by kd-trees. This approximation arises mainly from (i) simplifying the circular shape of cells to squares during the forward-translation step and (ii) discretizing continuous movements from the motion tensor during the back-translation step. Additionally, both the DORA and

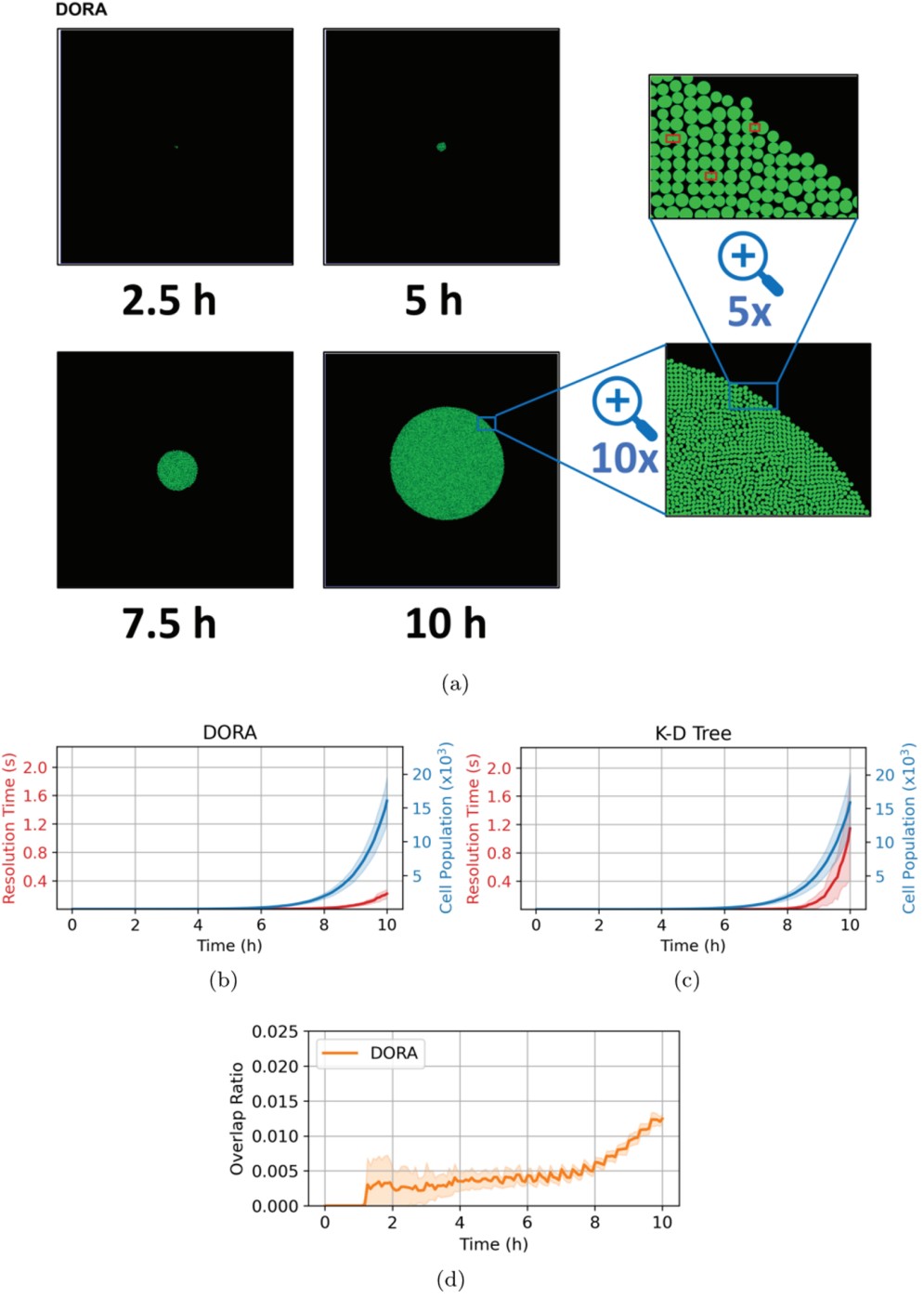

**Fig 4. Simulation of colony growth at a high nutrient concentration of** $100$ **mg/L, with overlap resolution managed by the DORA algorithm.** (**A**) Shows snapshots of the simulation at various time points, ending with a final population of $2 \times 10^4$ cells, and includes a close-up of the final colony illustrating DORA's effective overlap resolution; however, some minor residual overlaps remain, which are highlighted by rectangles. (**B**) Depicts the evolution of the resolution time step (in blue) and the cell population growth (in red) over the course of the experiment using the DORA algorithm. (**C**) Depicts the same relationship as (b) but with the overlap resolution occurring via the kd-tree approach for comparison. (**D**) Displays the progression of the overlap ratio during the simulation using the DORA algorithm. For all the plots, a line represents the mean of the results across 30 simulations while the thickness of a line represents the standard deviation.

the kd-tree methods operate under the assumption that cells are rigid and non-deformable upon contact which simplifies the interaction dynamics but does not account for potential cell deformations that occur in natural settings. These approximations can lead to minor residual overlaps, observable in Fig 4A. To assess the significance of these approximations, the residual overlap area was measured relative to the total colony area, as shown in Fig 4D. The overlap ratio remains below 1% for most of the simulation, with a slight increase noted during the final stages of rapid colony expansion. Reducing the interval of slow simulation timesteps can potentially lower the overlap ratio by allowing more frequent updates in later stages. For kd-tree-based approaches, periodic tree rebalancing is necessary to sustain efficiency during rapid population growth.

## Fractal growth

In this section, we explore the DORA algorithm's ability to simulate the fractal growth patterns characteristic of microbial colonies under nutrient-limited conditions. In such environments, the intense competition for scarce resources often results in the emergence of fractal structures along the colony edges. These structures enhance the fractal dimension of the edges, thereby increasing the colony's ability to exploit the limited available nutrients. Two distinct simulation sets were conducted with nutrient concentrations set to 10 mg/L and 1 mg/L, respectively. These conditions are expected to lead to diminished growth rates, reduced overall population densities, and the emergence of distinctive fractal growth patterns [19], [14].

Figs 5A, 5B, 6A, and 6B depict the temporal development of microbial colonies simulated by both the DORA and kd-tree algorithms at nutrient concentrations of 10 mg/L and 1 mg/L, respectively. In both scenarios, fractal growth is evident, with the key difference being that at 1 mg/L, fractal patterns emerge earlier, and overall growth is slower compared to the 10 mg/L case. This observation aligns with the expectation that severe nutrient scarcity exacerbates the competition for resources, thereby amplifying fractal branching and slowing colony expansion. In both cases, the DORA algorithm effectively captures the fractal growth patterns characteristic of nutrient-limited environments, closely aligning with the results generated by the kd-tree approach.

The computational performance of the DORA algorithm, compared to the kd-tree method, is illustrated in Figs 5C and 6C. While DORA maintains superior computational efficiency relative to the kd-tree approach across both nutrient conditions, its relative advantage narrows, especially at the 1 mg/L concentration. This decrease in relative efficiency can be attributed to two main factors: a decrease in the number of neighbor calculations per node in the kd-tree owing to less dense growth at lower nutrient levels and a marginal reduction in DORA's efficiency in scenarios with lower cell densities, where the algorithm processes a higher volume of empty grid spaces, thereby increasing the computational burden.

Additionally, the overlap ratios evolution in time presented in Figs 5D and 6D demonstrate the DORA algorithm's success in maintaining low overlap ratios, consistently remaining below 1% in both sets of simulations. Notably, an intermediate peak in the overlap ratio is observed between 5 to 10 hours in Fig 5D. This peak coincides with the period of exponential growth at the beginning of the simulation, where the rapid addition of new cells leads to a temporary increase in overlaps. As the growth rate slows due to substrate depletion, the overlap resolution becomes more manageable, resulting in a decrease in the overlap ratio. This highlights the DORA algorithm's ability to accurately model realistic microbial growth patterns across various nutrient conditions.

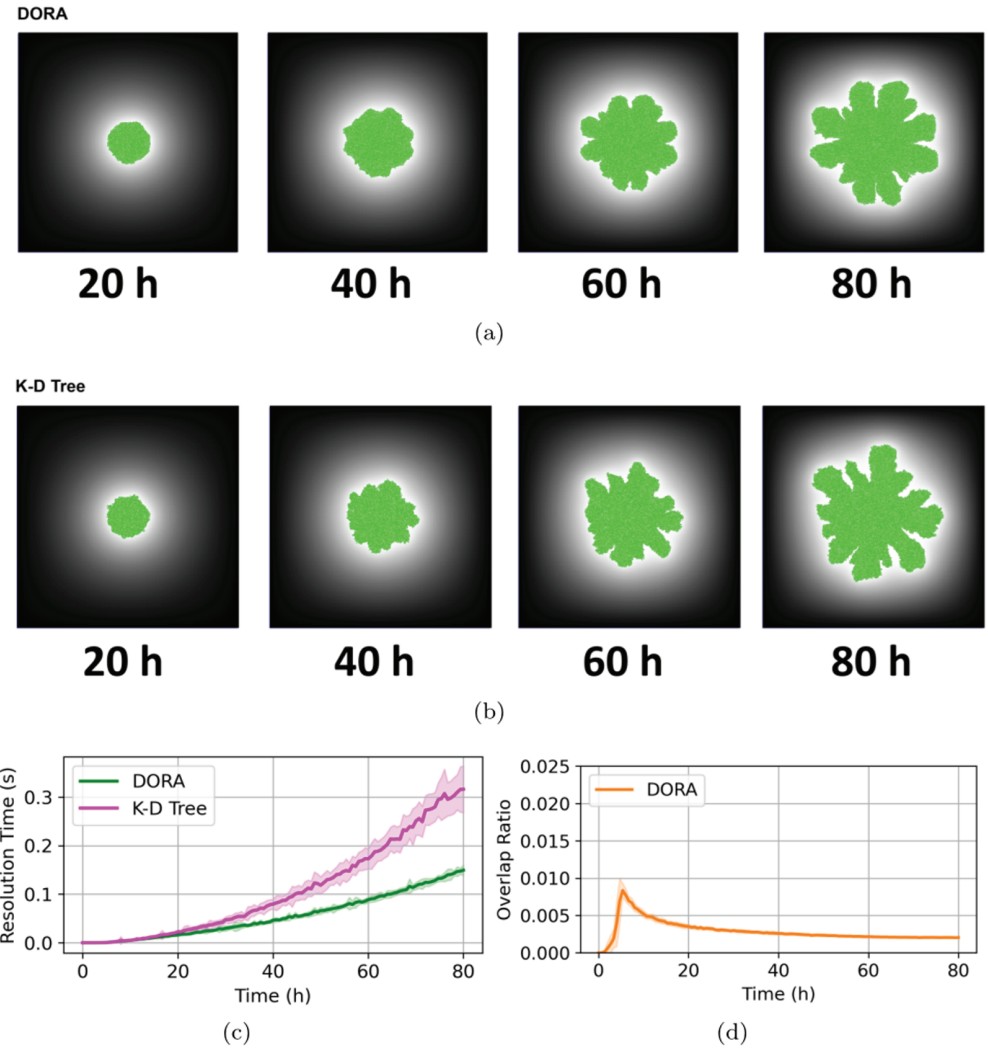

**Fig 5. Simulation of colony growth at a medium nutrient concentration of 10 mg/L.** (**A**) Presents snapshots of the simulation at various time points using the DORA algorithm. (**B**) Displays snapshots from the simulation at different time points while using the kd-tree approach. (**C**) Compares the resolution time step at each iteration of the simulation (on the y-axis) with the experiment time (on the x-axis) for DORA (in green) and kd-tree (in magenta). (**D**) Shows the evolution of the overlap ratio over time when using the DORA algorithm.

## Biofilm growth

In our final series of experiments, we sought to model biofilm formation through both the DORA and kd-tree algorithms across varying nutrient gradients. Biofilm development under high nutrient concentrations typically results in uniform growth patterns, as abundant resources minimize competition and allow for even expansion. Conversely, as nutrient availability decreases, biofilms begin to adopt more complex, mushroom-like structures, eventually resulting in pronounced fractal patterns under conditions of severe nutrient scarcity [8]. Both algorithms successfully replicated these expected spatial patterns, with DORA enhancing computational efficiency in nutrient-rich environments, as evidenced in Fig 7. The computational demands of kd-trees increase significantly in high-density scenarios, particularly

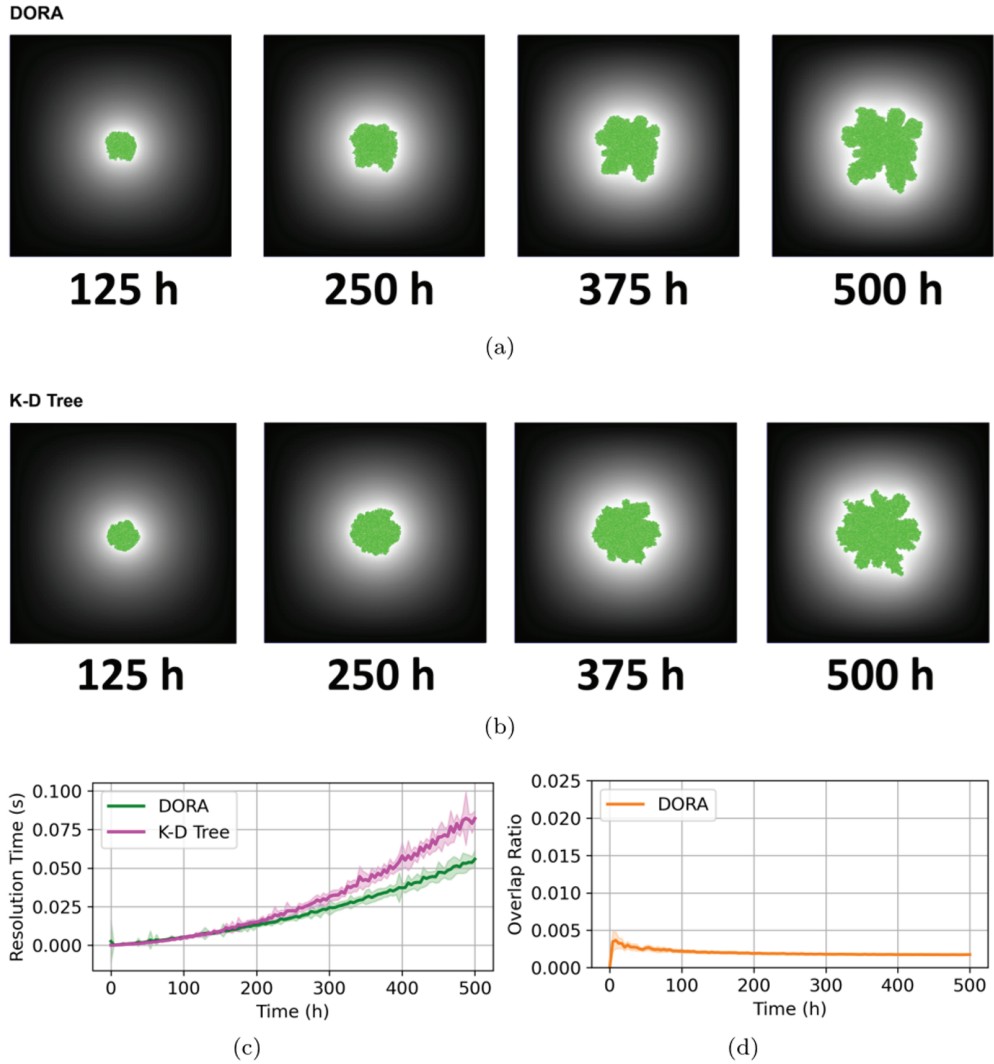

**Fig 6. Simulation of colony growth at a scarce nutrient concentration of 1 mg/L.** (**A**) Shows snapshots of the simulation at various time points using the DORA algorithm. (**B**) Displays snapshots from the simulation at different time points using the kd-tree approach. (**C**) Compares the resolution time step at each iteration of the simulation (on the y-axis) with the experiment time (on the x-axis) for DORA (in green) and the kd-tree (in magenta). (**D**) Shows the evolution of the overlap ratio over time when using the DORA algorithm.

when cell populations approach the $10^4$ magnitude. However, under nutrient-limited conditions, the performance gap between DORA and kd-tree diminishes, as depicted in Figs 8C and 9C. Despite the varying nutrient levels, both algorithms effectively captured the transition from uniform to mushroom-like, and eventually to fractal structures characteristic of biofilms under nutrient constraints, as illustrated in Figs 8 and 9.

Particularly notable are the biofilm simulations at a low nutrient concentration of 1 mg/L, as shown in Figs 9A and 9B. These simulations demonstrated that biofilms modeled by DORA presented a more consistent spatial structure compared to those generated by the kd-tree method. Although both algorithms facilitated the emergence of branching fractal structures under nutrient scarcity, a detailed examination reveals that the branch heights

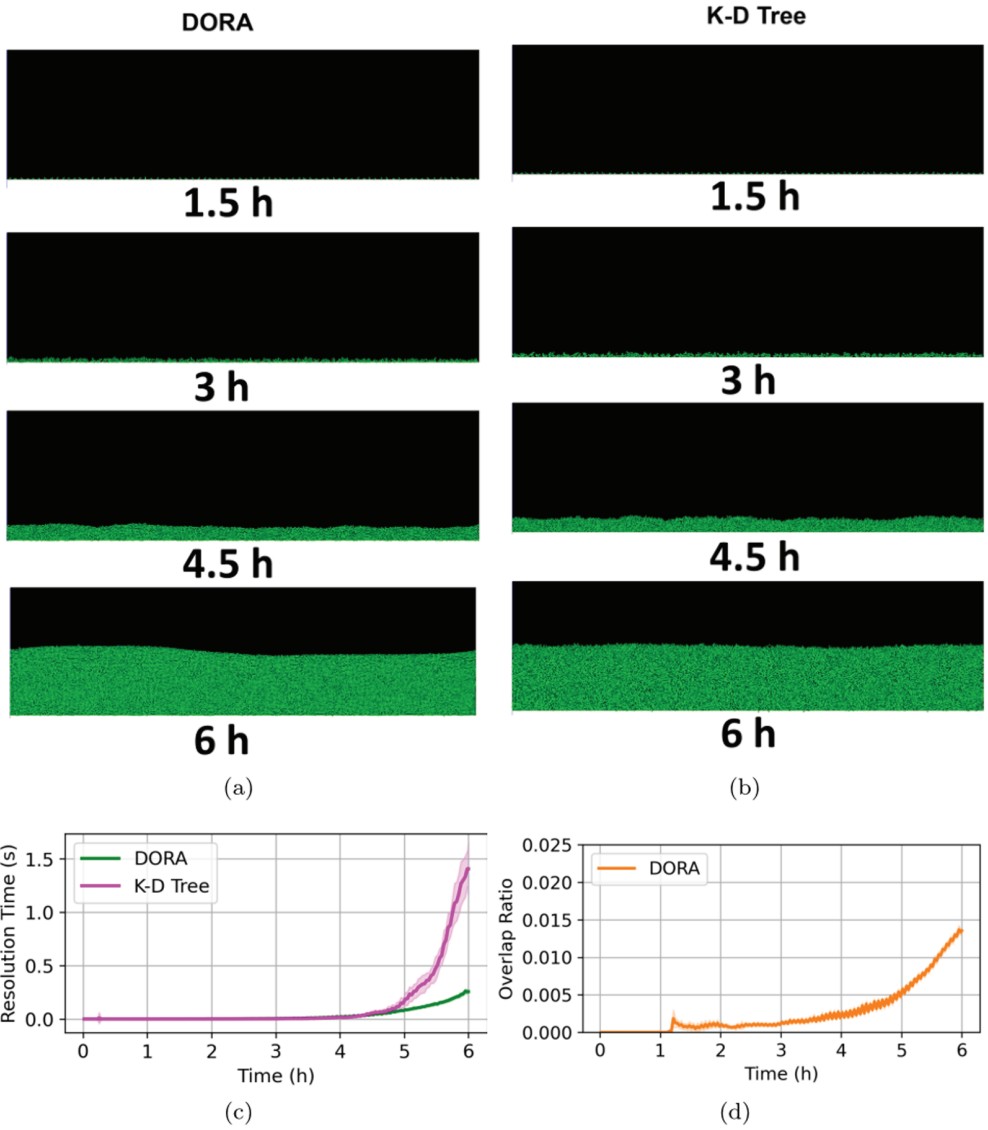

**Fig 7. Simulation of biofilm growth at a high nutrient concentration of 100 mg/L.** (**A**) Snapshots of biofilm development at different time points using the DORA algorithm. (**B**) Similar snapshots using the kd-tree approach. (**C**) Comparison the resolution time step at each iteration of the simulation for DORA (in green) and kd-tree (in magenta) against the experiment time. (**D**) Evolution of the overlap ratio over time when using the DORA algorithm.

in DORA-generated biofilms (Fig 9A) are more uniform than those in kd-tree-generated biofilms (Fig 9B). This consistency is likely a result of DORA's deterministic algorithmic approach, which reduces stochastic variability in the simulation process. In contrast, the kd-tree method's reliance on random neighbor processing introduces a higher degree of randomness, leading to a wider variety of biofilm architectures. It is crucial to acknowledge, however, that the overlap resolution algorithm is only one factor contributing to randomness within an IbM. Cellular processes such as metabolism, replication, lysis, and growth inherent to an IbM also contain inherent stochasticity. The choice of incorporating stochastic elements into

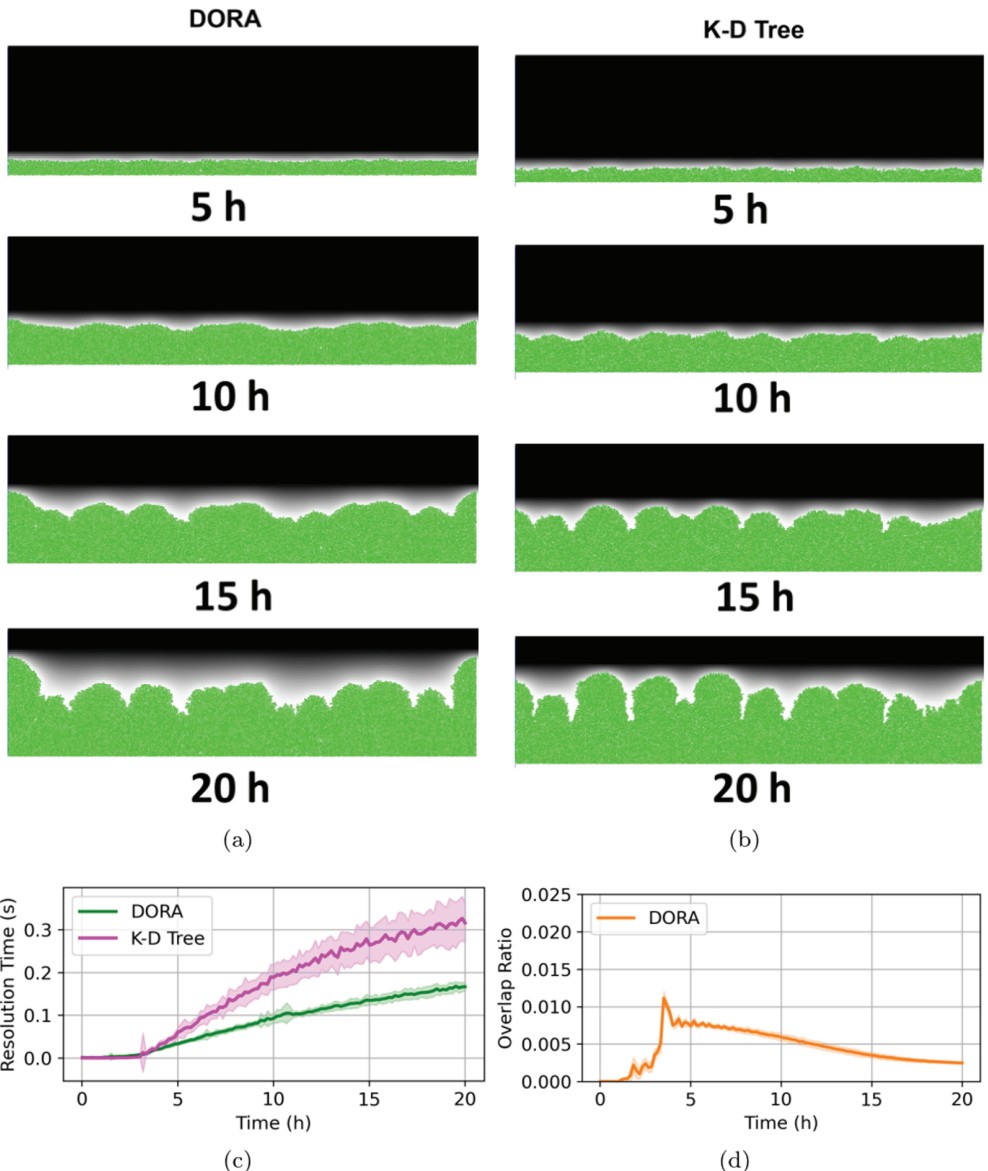

**Fig 8. Simulation of biofilm growth at an intermediate nutrient concentration of 10 mg/L.** (**A**) and (**B**) Biofilm development using DORA and kd-tree approaches, respectively, at various time points. (**C**) Comparison of the time required for overlap resolution in each iteration of the simulation between DORA (green) and kd-tree (magenta). (**D**) Evolution of the overlap ratio using DORA over the simulation duration.

the overlap resolution algorithm, a fundamentally deterministic physical process, remains ultimately a design choice of the modeler.

## Conclusion

This paper introduces DORA, a novel algorithm designed to address a significant computational bottleneck in the individual-based modeling of microbial growth: the resolution of spatial overlaps between cells. Our analysis demonstrates that DORA offers superior computational efficiency compared to traditional methods, particularly in simulations with dense

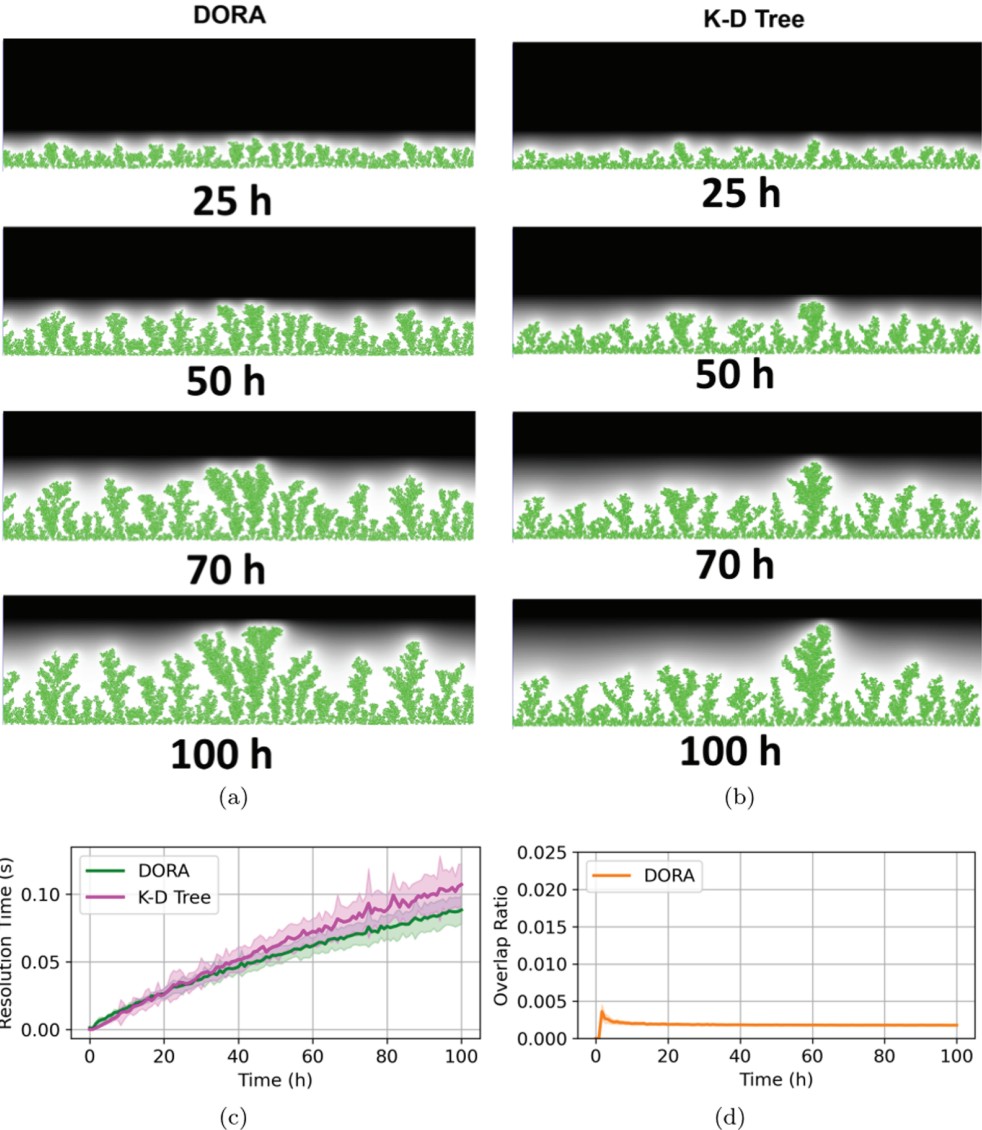

**Fig 9. Simulation of biofilm growth at a low nutrient concentration of 1 mg/L.** (**A**) and (**B**) Depiction of the progression of biofilm formation using the DORA and kd-tree methods, respectively. (**C**) Comparison of the time required of each algorithm in resolving overlaps during each simulation iteration, with DORA shown in green and kd-tree in magenta. (**D**) Overlap ratio evolution during the simulation with DORA.

microbial populations where conventional approaches, such as the kd-tree method, become computationally prohibitive. Furthermore, DORA retains high spatial accuracy in modeling microbial growth, even under nutrient-scarce conditions that typically give rise to complex spatial patterns. The algorithm's ability to accurately replicate these structures offers a clear advantage over simpler cellular automata models.

While we have demonstrated DORA's advantages over a kd-tree-based approach, its performance relative to other off-lattice agent-based modeling techniques, such as the particle Lenia approach [20], provides an interesting contrast. A DORA-based IbM framework offers a more direct and biologically intuitive way to model cellular processes, as it explicitly tracks

individual cells and their interactions within a discrete spatial environment, providing a direct representation of their biological processes. In contrast, particle Lenia relies on complex ODE formulations to approximate these interactions, which can be challenging to parameterize and may obscure mechanistic insights in systems where precise biological interactions need to be represented.

The particle Lenia approach employs a continuous spatial framework that allows for the flexible application of ordinary differential equations (ODEs) to model interactions and resolve overlaps between particles. This flexibility enables particle Lenia to incorporate stochastic elements effectively and is particularly suitable for simulating systems where randomness plays a critical role, such as microbial motility, diffusion processes, and environmental fluctuations.

DORA, in contrast, systematically resolves spatial overlaps using fixed rules within a discretized grid framework. This deterministic structure can be extended to include stochastic movements by applying a random displacement vector to cells before overlap resolution (see S2 Algorithm). This extension introduces randomness into cell movements, enabling simulations that reflect inherent biological variability while retaining computational efficiency.

Despite the flexibility and computational efficiency of particle Lenia, significant challenges remain in applying it to biological systems. A key difficulty lies in translating complex biological phenomena into the parameters required by particle Lenia. Formulating ODEs that faithfully represent the dynamics of biological interactions demands a deep understanding of the underlying mechanisms and requires fine calibration of these equations. While continuous, flexible parameter spaces in particle Lenia offer great versatility, they may oversimplify biological processes, potentially obscuring mechanistic understanding. This becomes particularly problematic when specific biological interactions or constraints need to be explicitly represented [23,24].

The benefits of DORA are twofold: firstly, it enables the simulation of densely populated microbial communities within manageable computational timeframes and with reduced memory requirements, extending the applicability of IbMs to scenarios previously considered challenging, such as biofilm formation in bioreactors. Secondly, the grid-based approach of DORA marks a significant step in bridging the gap between IbMs and their potential approximations via Partial Differential Equation (PDEs) models. While the description of many microbial growth processes within IbMs, including Monod growth kinetics, cellular replication, and nutrient diffusion, naturally lends itself to representation via PDEs, accurately approximating the shoving algorithms powered by exact kd-tree based methods for resolving spatial overlaps through differential equations remains a substantial challenge [25,26]. Consequently, future work will focus on refining and expanding DORA's grid-based methodology, exploring its integration with differential equation models to enhance the approximation of IbMs and contribute to a more unified framework for microbial growth simulation.

## Supporting information

**S1 Algorithm. Algorithm implementation using the Moore neighborhood.**
(PDF)

**S1 Table. Parameter values used in simulations.**
(PDF)

**S2 Algorithm. Algorithm implementation with stochastic motion.**
(PDF)

## Acknowledgments

We thank Lina Hashem for reviewing and refining parts of this manuscript.

## Author contributions

**Conceptualization:** Ihab Hashem.

**Formal analysis:** Ihab Hashem.

**Funding acquisition:** Jan F M Van Impe.

**Methodology:** Ihab Hashem.

**Project administration:** Jan F M Van Impe.

**Resources:** Jan F M Van Impe.

**Software:** Ihab Hashem, Jian Wang.

**Supervision:** Jan F M Van Impe.

**Visualization:** Jian Wang.

**Writing – original draft:** Ihab Hashem, Jian Wang, Jan F M Van Impe.

**Writing – review & editing:** Ihab Hashem, Jian Wang, Jan F M Van Impe.

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
