## [Decision Letter · Decision Letter 0]

5 Aug 2024

Dear Dr. Van Impe,

Thank you very much for submitting your manuscript "A Discretized Overlap Resolution Algorithm (DORA) for Resolving Spatial Overlaps in Individual-based Models of Microbes" for consideration at PLOS Computational Biology.

As with all papers reviewed by the journal, your manuscript was reviewed by members of the editorial board and by several independent reviewers. In light of the reviews (below this email), we would like to invite the resubmission of a significantly-revised version that takes into account the reviewers' comments.

We cannot make any decision about publication until we have seen the revised manuscript and your response to the reviewers' comments. Your revised manuscript is also likely to be sent to reviewers for further evaluation.

Sincerely,

David Basanta Gutierrez

Academic Editor

PLOS Computational Biology

Pedro Mendes

Section Editor

PLOS Computational Biology

Reviewer's Responses to Questions

**Comments to the Authors:**

Reviewer #1: Summary

The manuscript introduces the Discretized Overlap Resolution Algorithm (DORA) aimed at resolving cellular overlaps in individual-based models (IBMs) of microbial growth. DORA uses a grid-based framework to manage spatial overlaps, reducing computational costs compared to traditional pairwise comparison methods. The authors evaluate DORA's performance in simulating microbial colonies and biofilms under different nutrient conditions, demonstrating its computational efficiency and accuracy.

Major Comments

The manuscript primarily focuses on a deterministic approach. Consideration of stochastic elements, which are often relevant in microbial modeling, would broaden the applicability of the algorithm. The current approach is deterministic. How would the algorithm adapt to stochastic IBMs, where random elements influence cell behavior and interactions? The algorithm handles horizontal and vertical movements. Diagonal movements should also be considered to fully capture the spatial dynamics of microbial growth.

The manuscript lacks a comparative analysis with other off-lattice agent-based modeling techniques or the particle Lenia approach. These methods allow for arbitrary ODEs to adjust particle overlap, potentially offering more flexibility. Highlighting the advantages and limitations of DORA relative to these approaches would strengthen the manuscript. A detailed discussion of the superiorities of DORA compared to other existing methods (other than kd-tree) would be beneficial. This includes scenarios where DORA excels and where it might be limited.

The description of the algorithm is clear but could benefit from more illustrative examples or pseudocode to guide readers through the steps, especially those less familiar with computational modeling.

Minor Comments

On page 2, the manuscript should refer to Figure 1, not Figure 3. Please ensure accurate figure references.

Ensure consistent use of terminology throughout the manuscript. For example, the terms "cell" and "bacterial cell" are used interchangeably; choose one term for consistency.

Reviewer #2: The review is uploaded as an attachment

**Have the authors made all data and (if applicable) computational code underlying the findings in their manuscript fully available?**

Reviewer #1: None

Reviewer #2: **No: **The code used and data generated have not been made fully available by the authors yet

PLOS authors have the option to publish the peer review history of their article (what does this mean?). If published, this will include your full peer review and any attached files.

Reviewer #1: No

Reviewer #2: No
---

## [Decision Letter · Decision Letter 1]

10 Dec 2024

PCOMPBIOL-D-24-00908R1

A Discretized Overlap Resolution Algorithm (DORA) for Resolving Spatial Overlaps in Individual-based Models of Microbes

PLOS Computational Biology

Dear Dr. Van Impe,

Thank you for submitting your manuscript to PLOS Computational Biology. After careful consideration, we feel that it has merit but does not fully meet PLOS Computational Biology's publication criteria as it currently stands. Therefore, we invite you to submit a revised version of the manuscript that addresses the points raised during the review process.

Please submit your revised manuscript within 30 days Feb 09 2025 11:59PM. If you will need more time than this to complete your revisions, please reply to this message or contact the journal office at ploscompbiol@plos.org. Please include the following items when submitting your revised manuscript:

We look forward to receiving your revised manuscript.

Kind regards,

David Basanta Gutierrez

Academic Editor

PLOS Computational Biology

Pedro Mendes

Section Editor

PLOS Computational Biology

Feilim Mac Gabhann

Editor-in-Chief

PLOS Computational Biology

Jason Papin

Editor-in-Chief

PLOS Computational Biology

**Comments to the Authors:**

**Please note that the review is uploaded as an attachment.**

**Reviewers' comments:**

Reviewer's Responses to Questions

Reviewer #1: Thanks for considering my comments and your responses.

Comment 1.1

The response to the first comment provides a theoretical explanation of how their deterministic approach (DORA) could be adapted to integrate stochastic elements, but without actual simulations or experimental validation, it remains speculative. Anyway,

thank you for considering the inclusion of stochastic elements in DORA. However, the response would benefit from empirical evidence, such as proof-of-concept simulations or examples demonstrating how DORA can integrate with stochastic IBMs. This would provide stronger support for the claims and highlight the practical adaptability of the algorithm. The addition of a random motion vector is an intriguing idea, but the manuscript would be strengthened by detailing how this would be implemented, its computational implications, and specific scenarios where this adaptation might be beneficial.

Comment 1.3

I’m not sure about DORA's unique contributions. While the theoretical comparison is helpful, incorporating empirical results (e.g., simulations) would better support the claims. Are there specific biological systems where DORA’s deterministic approach is particularly advantageous?

Reviewer #2: Thank you to the authors for addressing my comments in detail. They have provided a more complete version of their work, with modified figures that facilitate the communication. I am keen to recommend this manuscript for acceptance with some changes.

Major

1. The explanation of the methods has been clarified. However, I found the "Forward Translation" section still challenging to understand. In particular, the authors should consider giving a step-by-step introduction to Eq. (2) and modifying Fig 3(a) to locate the labels i, j, etc., next to the lines. This little detail in Fig 3(a) prevented me from understanding Eq (2) right away. At present, the notation i,j can be misinterpreted as referring to a grid cell enumeration, while in reality, it indicates the geometric location at the intersection of the lines shown in Fig 3(a).

Minor

2. Thank you for providing your code and data in an online repository. Please consider adding it to a permanent database such as Zenodo and include a license. See, for example, https://docs.github.com/en/repositories/archiving-a-github-repository/referencing-and-citing-content

and

https://docs.github.com/en/communities/setting-up-your-project-for-healthy-contributions/adding-a-license-to-a-repository

3. The explanation of how to construct a tree in Fig 1(a) helps. However, I suggest linking the spatial distribution show to the tree show, making the explanation even more apparent. If I interpreted your figure correctly, the current tree shows ABCDEFG but this does not relate to the spatial distribution shown above it.

4. Consider mentioning that DORA is used for circular cells in the Abstract. This information might be important information that readers expect to find early on.

5. There are typos in lines 247 and 255. The units g/mL, etc, should be replaced by mg/L.

**Have the authors made all data and (if applicable) computational code underlying the findings in their manuscript fully available?**

Reviewer #1: None

Reviewer #2: Yes

PLOS authors have the option to publish the peer review history of their article (what does this mean?). If published, this will include your full peer review and any attached files.

Reviewer #1: No

Reviewer #2: No

**Figure resubmission:**
---

## [Decision Letter · Decision Letter 2]

19 Mar 2025

Dear Dr. Van Impe,

We are pleased to inform you that your manuscript 'A Discretized Overlap Resolution Algorithm (DORA) for Resolving Spatial Overlaps in Individual-based Models of Microbes' has been provisionally accepted for publication in PLOS Computational Biology.

Best regards,

David Basanta Gutierrez

Academic Editor

PLOS Computational Biology

Pedro Mendes

Section Editor

PLOS Computational Biology

Reviewer's Responses to Questions

**Comments to the Authors:**

Reviewer #1: I think this updated version of the paper is good enough to publish.

Reviewer #2: Thank you to the authors for addressing all my comments in detail. In the updated manuscript, the methods explaining the DORA algorithm are precise and facilitate readability. Together with the changes to figures and uploading their code to a public repository, it enables replicability of results. Altogether, this submission meets the requirements for publication in PLoS Computational Biology.

**Have the authors made all data and (if applicable) computational code underlying the findings in their manuscript fully available?**

Reviewer #1: Yes

Reviewer #2: Yes

PLOS authors have the option to publish the peer review history of their article (what does this mean?). If published, this will include your full peer review and any attached files.

Reviewer #1: No

Reviewer #2: No

---

## [Editor Report · Acceptance letter]

PCOMPBIOL-D-24-00908R2

A Discretized Overlap Resolution Algorithm (DORA) for Resolving Spatial Overlaps in Individual-based Models of Microbes

Dear Dr Van Impe,

I am pleased to inform you that your manuscript has been formally accepted for publication in PLOS Computational Biology. Your manuscript is now with our production department and you will be notified of the publication date in due course.

With kind regards,

Dorothy Lannert
